# Bridging Critical Gaps in Convergent Learning: How Representational Alignment Evolves Across Layers, Training, and Distribution Shifts

**Chaitanya Kapoor, Sudhanshu Srivastava, Meenakshi Khosla**
Department of Cognitive Science
University of California, San Diego
La Jolla, CA 92093
{chkapoor, sus021, mkhosla}@ucsd.edu

## Abstract

Understanding convergent learning—the degree to which independently trained neural systems—whether multiple artificial networks or brains and models—arrive at similar internal representations—is crucial for both neuroscience and AI. Yet, the literature remains narrow in scope—typically examining just a handful of models with one dataset, relying on one alignment metric, and evaluating networks at a single post-training checkpoint. We present a large-scale audit of convergent learning, spanning dozens of vision models and thousands of layer-pair comparisons, to close these long-standing gaps. First, we pit three alignment families against one another—linear regression (affine-invariant), orthogonal Procrustes (rotation-/reflection-invariant), and permutation/soft-matching (unit-order-invariant). We find that orthogonal transformations align representations nearly as effectively as more flexible linear ones, and although permutation scores are lower, they significantly exceed chance, indicating a privileged representational basis. Tracking convergence throughout training further shows that nearly all eventual alignment crystallizes within the first epoch—well before accuracy plateaus—indicating it is largely driven by shared input statistics and architectural biases, not by the final task solution. Finally, when models are challenged with a battery of out-of-distribution images, early layers remain tightly aligned, whereas deeper layers diverge in proportion to the distribution shift. These findings fill critical gaps in our understanding of representational convergence, with implications for neuroscience and AI.

## 1 Introduction

Deep Neural Networks (DNNs) are becoming increasingly popular in neuroscience for predicting neural responses [48, 47, 28], or as models for reverse-engineering algorithms of neural computation [42, 43, 11]. This congruence invokes the necessity to gain a deep understanding of how DNNs learn to represent information. A core question in this domain is whether independently trained networks converge on similar internal representations—and if so, under what conditions and along which dimensions this convergence unfolds? Comparative analysis of model representations helps reverse engineering neural networks by linking architectural components, training objectives, and data inputs to learned representations and, in turn, model behavior.

Over the past decade, there has been growing recognition that similar representations emerge across diverse models, despite differing in architecture, training procedures, or data modalities. Early work

---

All code is publicly available at: https://github.com/NeuroML-Lab/representation-alignment

39th Conference on Neural Information Processing Systems (NeurIPS 2025).

demonstrated that independent training runs of the same architecture develop a core set of features that align well across networks. For example, early layers in convolutional networks learn Gabor-like filters, across a range of architectures and tasks [49, 32]. Efforts to quantify these similarities have employed techniques such as canonical correlation analysis (CCA) and its variants [36, 39], centered kernel alignment (CKA) [31], representational similarity analysis (RSA) [35] and model stitching [4]. These studies have underscored the high degree of alignment in early layers and noted convergence even in later stages of the network. These findings provide empirical grounding for theories of representational convergence, hinting at the existence of universal principles governing learning, which may also shed light on how biological neural circuits process information.

Recent studies have extended this line of inquiry, demonstrating that increasing model capability—through increased scale, multitask training, or cross-modal learning—drives representational convergence not just within, but also *across* modalities [37]. The Platonic Representation Hypothesis further argues that as models scale and solve more tasks, they are driven to discover a universal, modality-agnostic representation of reality [26]. Yet, identifying precise conditions of network convergence on similar representations and its implications remain open questions.

To address these gaps, in our work, we examine representational alignment along three key axes:

**Across layers:** Prior work has shown that early layers tend to extract general, low-level features—such as edge detectors in vision networks—while deeper layers develop task-specific representations. Studies [31, 35, 34] have quantified these changes using methods like CCA and CKA. However, these approaches often rely on single metrics that obscure minimal transformations needed to align representations. Understanding the precise nature of these transformations is critical for dissecting how representations in different networks relate to each other (*e.g.*, are they similar in information content, representational geometry, or even at the level of single-neuron tuning?). It is also unknown whether hierarchical (layer-wise) correspondence holds for other metrics with more restricted invariances than affine transformations.

**Across training:** A second key question is *when* convergence emerges. Most studies snapshot networks only after training has finished, but revealing the mechanisms behind alignment demands tracking how representations co-evolve during training, not just examining the final state. Conventionally, it is assumed that as different networks optimize on a task, their internal representations become more similar, driven by the final task solution. This assumption is the basis for the *contravariance principle* [9], which posits that when a network is pushed to achieve a difficult task, there is less room for variation in the final solution, forcing representations to converge. However, the question of when representational convergence occurs *during* training remains underexplored. Understanding this dynamic can illuminate the roles of initialization, early data statistics, architectural biases, learning dynamics, and the final task solution in shaping alignment. Previous studies have shown a *"lower layers learn first"* [39] behavior by comparing layers over time on CIFAR-10 using SVCCA, but little is known about the dynamics of convergence, especially for complex vision networks and using other metrics. More recently, Atanasov et al. [3] described a *"silent alignment effect"* in which a network's output aligns with the target function early in training well before the loss falls. Theoretical studies such as [8, 15, 41] present an analytical framework for the temporal dynamics of learning in deep *linear* networks, distinguishing an early *"lazy"* phase, characterized by kernel regression dynamics [27]—from a subsequent *"rich"* phase characterized by non-linear feature learning. While these models provide valuable insights into training dynamics and convergence in linear regimes, they fall short of fully explaining representational alignment in non-linear architectures. Though deep linear models provide valuable theoretical traction, further work is needed to elucidate how representational alignment develops and generalizes to the rich, non-linear regimes typical of contemporary DNNs.

**Across distribution shifts:** Although many DNNs exhibit highly human-like responses to in-distribution stimuli, there is mounting evidence that their responses can diverge dramatically under out-of-distribution (OOD) conditions [38, 20, 22]. Despite this, the effect of OOD inputs on model-to-model representational convergence remains poorly understood. Exploring this axis is crucial for rigorously testing the universality of learned representations.

**Key contributions.** In this work, we address these critical gaps by performing a large-scale systematic audit of representational convergence along these three axes. First, we employ three alignment metrics—linear regression [48, 42, 51, 33], Procrustes analysis [45, 46], and permutation-based methods [34, 29, 30]—each with different restrictions on the freedom of the mapping function, to identify the minimal set of transformations needed to align representations across networks reasonably well for each layer. This approach allows us to dissect how representations relate to each other, whether in terms of information content, representational geometry, or single-neuron tuning. Second, we examine the temporal dynamics of convergence during training, revealing that nearly all alignment occurs within the first epoch, challenging the assumption that convergence is tied to task-specific learning. Finally, we explore how changes in input statistics affect representational alignment across layers, demonstrating how OOD inputs differentially affect later versus early layers.

## 2 Problem Statement

We consider two representations $\boldsymbol{X}_i \in \mathbb{R}^{M \times N_x}$ and $\boldsymbol{X}_j \in \mathbb{R}^{M \times N_y}$ obtained from different models over $M$ unique stimuli, where $N_x$ and $N_y$ denote the number of neurons (or units) in each representation, respectively. To systematically identify the minimal transformations needed for alignment, we use three metrics that quantify similarities between networks while ignoring nuisance transformations. Among these are **(a)** the **linear regression score** which discounts affine transformations, **(b)** the **Procrustes score** which discounts rotations and reflections, treating them as nuisance factors and **(c)** the **permutation score** which considers the order of units in the representations as arbitrary. These metrics are ordered to reflect progressively less permissive mapping functions (from flexible to strict). When representations have an equal number of neurons ($N_x = N_y = N$), each of these similarity metrics seeks to minimize their Euclidean distance by optimizing over a set of $N \times N$-dimensional mapping matrices $\boldsymbol{M}$, $\min_{\boldsymbol{M}} \|\boldsymbol{X}_i - \boldsymbol{M}\boldsymbol{X}_j\|_2^2$. Linear regression score imposes no constraint on $\boldsymbol{M}$, Procrustes score enforces the matrices to be orthogonal (i.e. $\boldsymbol{M} \in \mathbf{O}(N)$) and permutation score requires the matrices to be permutation matrices ($\boldsymbol{W} \in \mathcal{P}(N)$)). When $N_x \neq N_y$, we use a generalized version of permutation-based alignment called the soft-matching score [29]. Here, the mapping matrix $\boldsymbol{M} \in \mathbb{R}^{N_x \times N_y}$ is constrained such that its entries are nonnegative and satisfy $\sum_{j=1}^{N_y} \boldsymbol{M}_{ij} = \frac{1}{N_x} \quad \forall i = 1, \ldots, N_x, \sum_{i=1}^{N_x} \boldsymbol{M}_{ij} = \frac{1}{N_y} \quad \forall j = 1, \ldots, N_y$. These constraints place $\boldsymbol{M}$ in the transportation polytope $\mathcal{T}(N_x, N_y)$ [13].

Once the optimal mapping matrix $\boldsymbol{M}$ is computed for each metric, we report the alignment using the pairwise correlation $\texttt{Alignment} = \text{corr}\,(\boldsymbol{X}_j,\ \boldsymbol{M}\boldsymbol{X}_i)$. For asymmetric metrics, we report the average alignment score computed in both directions: $\text{corr}(\boldsymbol{X}_j,\ \boldsymbol{M}_1\boldsymbol{X}_i)$ and $\text{corr}(\boldsymbol{X}_i,\ \boldsymbol{M}_2\boldsymbol{X}_j)$, where $\boldsymbol{M}_1$ and $\boldsymbol{M}_2$ are the respective transformations. We also report our results using Spearman's rank correlation coefficient in Appendix A2 to address the possibility of being susceptible to high-variance activation dimensions.

By leveraging these metrics—which progressively relax mapping constraints from strict (soft-matching/permutation) to flexible (linear regression)—we can dissect the nature of representational alignment across networks, distinguishing between similarity in representational form (captured by the soft-matching score, which reduces to the permutation score when $N_x = N_y$), geometric shape (captured by the Procrustes score), and information content (captured by the linear regression score).

## 3 Method

Below, we outline our framework for evaluating alignment between different deep convolutional vision models (ResNet18, ResNet50 [25], VGG16, VGG19 [44]), with their training procedures described in Appendix A1. We demonstrate the robustness of all these results on CIFAR100 in Appendix A4.

### 3.1 Comparing Convolutional Layers

For a given convolutional layer, let the activations be represented by $\boldsymbol{X} \in \mathbb{R}^{m \times h \times w \times c}$, where $m$ is the number of stimuli, $h$ and $w$ denote the spatial height and width, and $c$ is the number of channels (i.e., convolutional filters). Each convolutional layer produces a feature map whose spatial dimensions are equivariant to translations. That is, a circular shift along the spatial dimensions yields an equivalent

representation (up to a shift). As a result, one could compare the full spatial activation patterns between networks by considering an equivalence relation that allows for spatial shifts. However, evaluating an alignment that optimizes over all possible shifts together with another alignment (*e.g.*, Procrustes) is computationally costly.

Previous work has shown that optimal spatial shift in convolutional layers tend to be close to zero [46]. This motivates our simpler approach: rather than collapsing the spatial dimensions by flattening the entire feature map (which would yield $X \in \mathbb{R}^{m \times (h \cdot w \cdot c)}$), we extract a single representative value from each channel. In our experiments, we choose the value at the center pixel of each channel, reducing the activation tensor $X$ to a two-dimensional matrix $X' \in \mathbb{R}^{m \times c}$, where each row corresponds to a stimulus and each column to a channel. This drastically reduces the computational complexity of computing the optimal mapping. For instance, aligning full, spatially-flattened representations would incur a runtime of $\mathcal{O}(mh^2w^2c + h^3w^3c^3)$, whereas our center-pixel approach reduces the problem to aligning an $m \times c$ matrix, resulting in a more tractable complexity.

## 3.2 Computing Alignment

Alignment was quantified in four regimes: (1) **Within–architecture:** each layer aligned with its counterpart in an independently initialized instance of the same architecture (trained on CIFAR100 or ImageNet); (2) **Across architectures:** all layer–layer alignments between different architectures; (3) **Across training:** analysis repeated after each of the first ten epochs; (4) **Under distribution shift:** within–architecture alignments tested on ImageNet-trained models using 17 OOD datasets from [20]. All scores are averaged over five-fold cross-validation ($k = 5$).

# 4 Results

## 4.1 Evolution of Convergence Across the Network Hierarchy

**How convergence varies with network depth.**   When comparing representational convergence across the network hierarchy for different seeds of the same architecture, we observe that convergence is strongest in the earliest layers and gradually diminishes in deeper layers (Fig. 1). This pattern is consistent across all three metrics and across networks trained on ImageNet. The high alignment in early layers likely arises because they capture fundamental, low-frequency features (*e.g.,* edges, corners, contrast) that are universal across representations [40, 5, 50]. In contrast, deeper layers, while still showing significant alignment ($> 0.5$), exhibit greater variability due to their sensitivity to specific training conditions and noise. We also contrast this result with randomly initialized (untrained) networks as a baseline. We find that alignment scores across all metrics are consistently lower in untrained networks compared to their trained counterparts. This difference is especially pronounced in deeper layers: for instance, using the Procrustes metric, the mean alignment increases by $145.26\%$ in early layers (depth $< 0.5$) and by $493.84\%$ in deeper layers (depth $> 0.5$) following training. This trend also holds when comparing networks with different architectures, underscoring the robustness of hierarchical convergence across diverse models. Interestingly, a similar hierarchical trend is observed in human brain responses to visual stimuli (Appendix A8).

**Minimal transformations needed to align representations.**   Across all layers, we find that alignment scores increase as the mapping functions become more permissive (Permutation → Procrustes → Linear), as expected. However, linear mappings provide only modest improvements over Procrustes correlations, indicating that rotational transformations are sufficient to capture the majority of alignment information. This suggests that the added flexibility of linear mappings—such as scaling and shearing—does not substantially enhance alignment beyond what is achieved with Procrustes transformations. Importantly, because Procrustes is symmetric, this result highlights that alignment reflects a deeper similarity in the geometric structure of representations, rather than merely the ability of one representation to predict another.

**Simple permutations achieve significant alignment.**   Despite the strict constraints imposed by permutation-based alignment, Permutation scores achieve surprisingly high alignment levels, indicating a strong one-to-one correspondence between individual neurons across network instances—suggesting that convergent learning extends down to the single-neuron level, even without allowing for more flexible transformations.

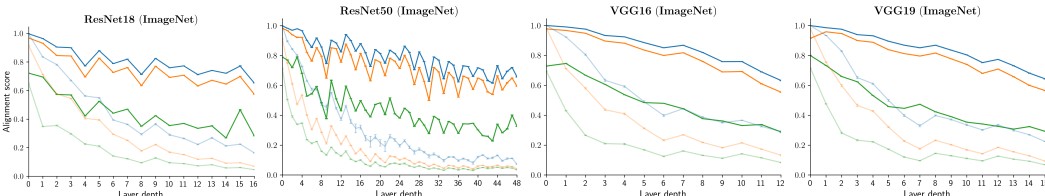

Figure 1: **Representational Convergence Across a Network Hierarchy.** We plot the evolution of alignment scores (computed between different seeds of the same network architecture) across the network hierarchy for four vision network architectures trained ImageNet. A consistent downward trend across layers indicates decreasing representational convergence as networks deepen. Alignment consistently follows the order: Linear > Procrustes > Permutation, reflecting the progressively stricter nature of the metrics. Lighter shades of the same color denote alignment for random networks. Notably, Procrustes transformations align representations nearly as well as Linear transformations, suggesting that most variability is due to rotations rather than more complex transformations. Even Permutation scores—despite their strictness—achieve substantial alignment, indicating a strong one-to-one correspondence between neurons across seeds, which points to stable, convergent neuron-level representations. Error bars represent the standard deviation computed across 5-fold cross-validation.

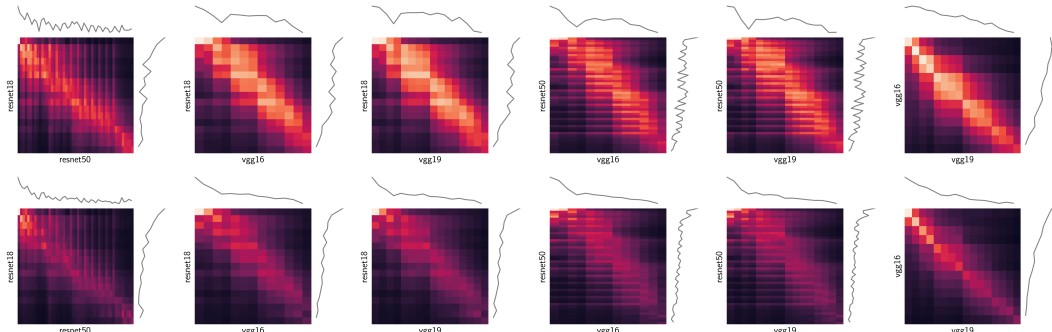

Figure 2: **Inter-Model Comparisons.** We consider all pairs of vision models, and for each pair, compute the alignment scores between every pair of layers using the orthogonal Procrustes (**Top**) and Soft-Matching (**Bottom**) metric trained on ImageNet. Gray line plots denote the **maximum** alignment value for each network over rows (right line) and columns (top line). A common trend that is observed here is the consistent relationships between layers of CNNs trained with different architectures.

To further probe this result and assess the depth of convergent learning, we tested the sensitivity of permutation alignment to changes in the representational basis.

Specifically, we applied a random rotation matrix $Q \in \mathbb{R}^{n \times n}$ to the converged basis of a neural representation, where $n$ is the number of neurons in a given layer. The rotation matrix was sampled from a Haar distribution via a $QR$ decomposition, ensuring that all orthogonal matrices were equally likely. We then recomputed the Permutation score after applying this rotation.

We conducted this analysis by taking response matrices from two identical DC-NNs (initialized with different random

| Model | Native (Min / Max) | Rotated (Min / Max) | Difference (%) (Min / Max) |
|---|---|---|---|
| ResNet18 | 0.202 / 0.721 | 0.154 / 0.581 | 10.47% / 201.84% |
| ResNet50 | 0.225 / 0.791 | 0.143 / 0.739 | 2.61% / 180.75% |
| VGG16 | 0.288 / 0.746 | 0.181 / 0.599 | 21.72% / 81.12% |
| VGG19 | 0.292 / 0.799 | 0.171 / 0.562 | 41.95% / 89.68% |

Table 1: **Sensitivity of Permutation Scores to Representational Axes.** For each ImageNet-trained network we apply a random rotation to the network's unit basis, recompute permutation-based alignment scores, and summarize the results across convolutional layers. Columns report the minimum and maximum alignment scores observed over layers in the native basis and in the rotated basis. The final column gives the percentage change in alignment after rotation. Rotations almost always reduce alignment, indicating the distinguished nature of the representational axes for all layers in networks.

seeds) at a given convolutional layer, $\{X_1, X_2\} \in \mathbb{R}^{m \times n}$, where $m$ represents the number of stimuli.

We applied the random rotation $Q$ to one network's responses and computed the resulting permutation-based correlation score, $s_{\mathrm{perm}}(X_1Q, X_2)$. This process was repeated across all convolutional layers, with the alignment differences summarized in Table 1.

These rotations consistently reduced alignment, with a drop between $\sim 10 - 202\%$ for ResNet18 across all layers, for instance. This significant decrease highlights that the learned representations are *not* rotationally invariant and the specific bases in which features are encoded is meaningfully preserved across networks. In other words, convergent learning aligns not just the overall representational structure but also the specific axes along which features are encoded. This observation echoes recent findings by [30], who report the existence of privileged axes in biological systems as well as the penultimate layer representations of trained artificial networks.

**Hierarchical correspondence holds across metrics.** Previous studies have shown that for architecturally identical networks trained from different initializations, the layer most similar to a given layer in another network is the corresponding architectural layer [31]. However, this finding has primarily been supported using metrics invariant to affine transformations (*e.g.,* CCA, SVCCA). Here, we extend this result by showing that stricter metrics—such as Procrustes and soft-matching scores—also reveal the same hierarchical correspondence (Fig. A3), even when comparing networks with different architectures. This suggests that the hierarchical alignment of representations is a fundamental property of neural networks, robust to both architectural differences and the choice of alignment metric. Moreover, our results show that both representational shape (captured by Procrustes) and neuron-level tuning (captured by soft-matching) follow similar alignment patterns, reinforcing the consistency of this hierarchical organization across different levels of representational analysis.

## 4.2 Evolution of convergence over training

We next explore how representational convergence evolves during the training process. Specifically, we compute Procrustes alignment scores between pairs of networks over training epochs. As shown

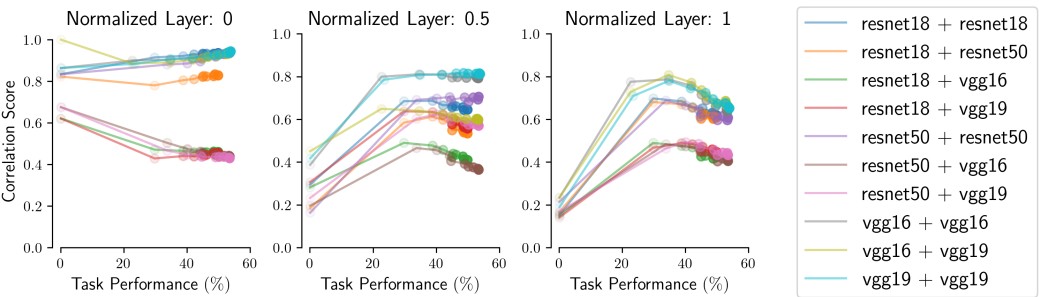

Figure 3: **Representational Alignment Through Training Evolution.** We visualize the evolution of Procrustes alignment between network pairs during task optimization on ImageNet. Lighter shades indicate earlier epochs, progressively darkening with later epochs. The plots span from epoch 0 (untrained) to epoch 10, with task performance improving over time. Epoch progression can be inferred from the increasing task performance along the $x$-axis.

in Fig. 3, a striking pattern emerges: the majority of representational convergence occurs within the first epoch—long before a networks task performance peaks. This rapid early convergence suggests that factors independent of task optimization drive much of the representational alignment (see also Appendix A7). Shared input statistics, architectural biases, and early training dynamics seem to play a dominant role in shaping representational convergence, overshadowing the influence of the final task-specific solution. This observation challenges prevailing hypotheses, such as the *contravariance principle* [9] and *task generality hypothesis* [26], which argue that networks align because only a narrow solution subspace yields high performance. Instead, our findings point to inductive biases and early learning processes as the dominant forces shaping representational convergence.

Our findings echo those of [17], who showed that most representational re-organization happens within the first few hundred iterations—well before any substantive task learning. Perturbing weights during this brief window (by re-initializing or shuffling them) severely impairs eventual accuracy,

indicating the formation of early, data-dependent structure. Together, these observations reinforce the view that shared input statistics—not task labels—are the principal drivers of early representational convergence.

Early convolutional layers show little training-induced change on representational similarity (even reducing in some cases). This phenomenon arises because early layers compute near-linear mappings of the input, even in untrained networks, allowing them to align well with simple linear transformations. Subsequent training only makes minor adjustments that slightly disrupt initial alignment, though they remain highly similar overall.

### 4.3 Convergence across distribution shifts

Having established alignment on in-distribution images, we next probed its robustness to distribution shift. We applied 17 out-of-distribution (OOD) variants from [21] to ImageNet-trained CNNs. Dataset specifics are given in Appendix A5.

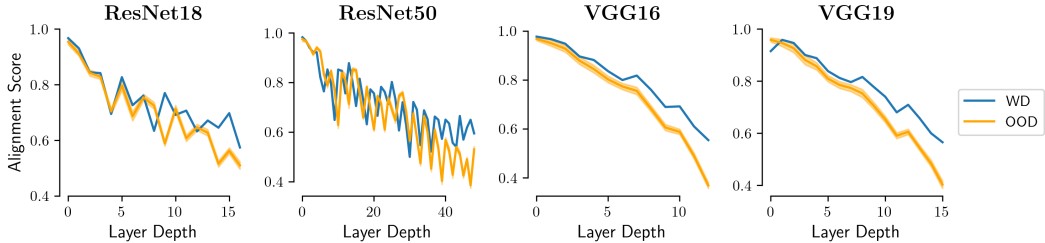

Figure 4: **Procrustes Score-based Alignment Between Networks Sharing the Same Architecture but Trained With Different Random Seeds, Plotted as a Function of Layer Depth.** Alignment is measured using within-distribution (WD) stimuli (ImageNet test set) and out-of-distribution (OOD) stimuli, with OOD values averaged across 17 datasets. Error bars represent the standard error computed across the $(n = 17)$ OOD datasets.

Using the Procrustes metric, we compute representational alignment on these datasets and observed a consistent pattern: OOD inputs amplify differences in the later layers of the networks, while early layers maintain comparable alignment levels between in-distribution and OOD stimuli (Fig. 4). We hypothesize that this pattern arises because early layers capture universal features (*e.g.,* edges, textures) that remain nearly identical across distributions, whereas later layers encode task-specific features that are sensitive to distributional shifts, thus amplifying the divergence between models. We also found a strong correlation between representational alignment and OOD accuracy: datasets where models maintained higher accuracy show strong alignment and vice versa. This correlation is minimal in early layers, but progressively increases with network depth across all architectures (Fig. 5); analogous trends appear in other networks Fig. A5.

These results have several important implications. First, early-layer alignment is remarkably stable—across random initializations and in-distribution and OOD inputs—indicating that these layers encode broadly transferable features that serve as a common scaffold for higher processing. Because this scaffold endures under distribution shift, one could plausibly improve OOD generalisation by fine-tuning only the later layers. Second, these findings inform model-brain comparisons. Though diverse architectures and learning objectives yield similar brain predictivity [12], the amplified divergence of later-layer representations under OOD conditions suggests that OOD stimuli could be especially useful to distinguish and select between models whose representations closely mirror the brain.

### 4.4 Representational Alignment for Self-Supervised Networks

We next test whether our findings generalize to self-supervised learning. All analyses were replicated on models trained with a Momentum Contrast (MoCo) objective [24] (training details in Appendix A1). (1) **Across hierarchy:** convergence is strongest in early layers and tapers with depth (Fig. 6-A); Linear and Procrustes scores remain comparable, suggesting inter-model variability reflects rotations/reflections. (2) **Across distribution shifts:** early layers maintain alignment for both in-distribution and OOD inputs, while deeper layers diverge (Fig. 6-B), indicating that self-

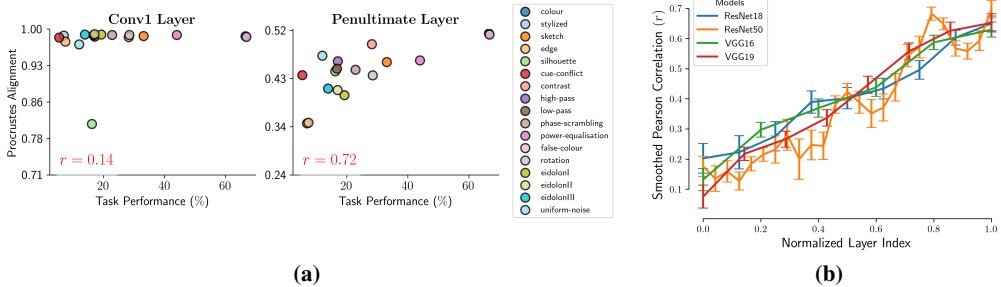

**(a)**                                                                                                     **(b)**

Figure 5: **Convergence on OOD Inputs. (a)** Procrustes alignment vs. task performance of ResNet50 models on each of the 17 datasets for the first convolutional layer **(Left)** and the penultimate **(Right)** layer. **(b)** Correlation between these variables as a function of network depth (normalized by each models depth).

supervised training, like supervised learning, produces stable, general early filters and increasingly distribution-sensitive higher representations. (3) **Across training:** most alignment emerges within the first epoch, well before optimal task performance (Fig. 6-C). (4) **Permutation sensitivity:** applying a random rotation substantially reduces permutation alignment (Fig. 6-D), confirming convergence to a privileged basis regardless of learning paradigm.

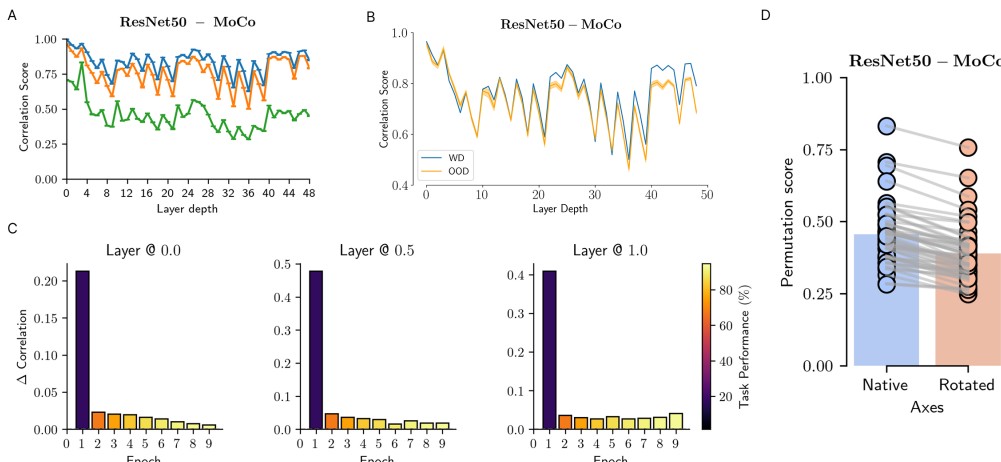

Figure 6: **Representational Alignment with Self-Supervised Networks. (A)** Representational alignment across layers of a pair of MoCo-trained ResNet50 models on ImageNet. Error bars indicate standard deviation from 5-fold cross-validation. **(B)** We plot the Procrustes alignment between MoCo models evaluated on in-distribution (ImageNet) and out-of-distribution (Stylized ImageNet [21]) stimuli. Error bars show standard error across all ($n = 17$) OOD datasets. **(C)** Change in Procrustes alignment score with every training epoch. Colors in the bar-plot indicate the top-1 accuracies during training. **(D)** Permutation alignment in the native and randomly rotated basis. Each dot corresponds to a convolutional ResNet50 layer. Bars indicate mean alignment. Rotation reduces mean alignment by $14.87\%$.

## 4.5    Representational Convergence in Vision Transformers

We next analyze representational convergence in ViTs. All models and training details are in Appendix A1. We analyze the [CLS] (class) token representations at each layer. This is because the [CLS] token aggregates the *"information content"* of all patch embeddings via the self-attention operator, making it a useful proxy to study representational geometry. In addition, computing representational similarity for [CLS] vectors avoids having to deal with pooling patch similarities into a single representative score.

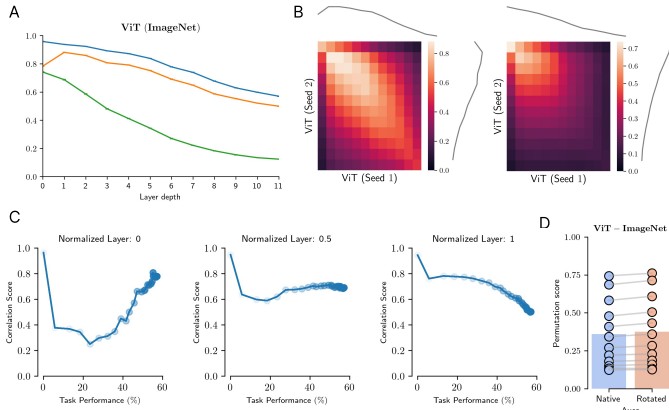

Figure 7: **Representational Convergence in Vision Transformers. (A)** We plot the evolution of alignment scores of three metrics (Linear, Procrustes, Permutation) computed between different seeds of the same ViT, which was trained on ImageNet. Error bars denote the standard deviation across 5-fold cross validation. **(B)** We plot the inter-model orthogonal Procrustes (**left**) and permutation (**right**) scores for all layer pairs. Gray line plots denote the maximum alignment value over rows (right line) and columns (top line). **(C)** We visualize the evolution of the orthogonal Procrustes score at different checkpoints, ranging from epochs 0 (untrained) to 25. Darker colors correspond to epoch progression. **(D)** We rotate the converged basis of ViTs by a random rotation matrix and recompute the permutation scores for models trained ImageNet. Each dot represents a layer in the ViT. The permutation alignment remains approximately constant in both cases.

**Results.** Analyses parallel to DCNNs reveal three key findings: (1) **Depth-wise convergence:** alignment tapers with network depth across metrics (Fig. 7-A); Procrustes and Linear scores remain comparable, indicating rotations/reflections explain most inter-seed variability, and layer-wise patterns mirror DCNNs (Fig. 7-B, left). (2) **No privileged axes:** independent ViT runs do not share a common basis—permutation scores between native and rotated axes are statistically indistinguishable across layers (Fig. 7-D; Appendix A3). (3) **Early plateau:** alignment stabilizes after the first epoch, except for a `[CLS]` embedding artifact due to uniform positional encoding, indicating late-stage training does not drive convergence (Fig. 7-C).

### 4.6 Representational Convergence in Language Models

We analyze representation convergence in language models using sentence embeddings from the Semantic Textual Similarity Benchmark (STSB) [10]. Our analyses include two primary comparisons: **(i) Same architecture:** multiple **Pythia-160m** instances differing only by random seed, allowing us to analyze variability in representational spaces solely from stochastic training factors **(ii) Cross-architecture: Pythia-70m** vs. **Pythia-160m**, to probe the effect of network depth on learned representations.

**Methodology.** For our representational similarity analysis, we perform two sets of computations: **(i) Final Checkpoint:** we compute the similarity between every pair of layers for all models at their final checkpoints using the same metrics as in the case of vision models (Linear, Procrustes, Permutation). **(ii) Intermediate Checkpoint:** across 154 intermediate checkpoints, we analyze layer representations at normalized depths 0 (beginning), 0.5 (middle), 1 (end) in the network. We track changes in the Procrustes similarity over training.

**Results.** Comparing representations across language models reveals three key patterns: (1) **Metric-dependent alignment:** alignment follows Linear > Procrustes > Permutation (Fig. 8-A); Procrustes and Linear scores are nearly identical, indicating that rotations/reflections explain most variability across models, and high alignment persists across layers. (2) **Hierarchical correspondence:** layers at similar depths align most closely both within (Fig. 8-B) and across architectures (Fig. 8-C). (3)

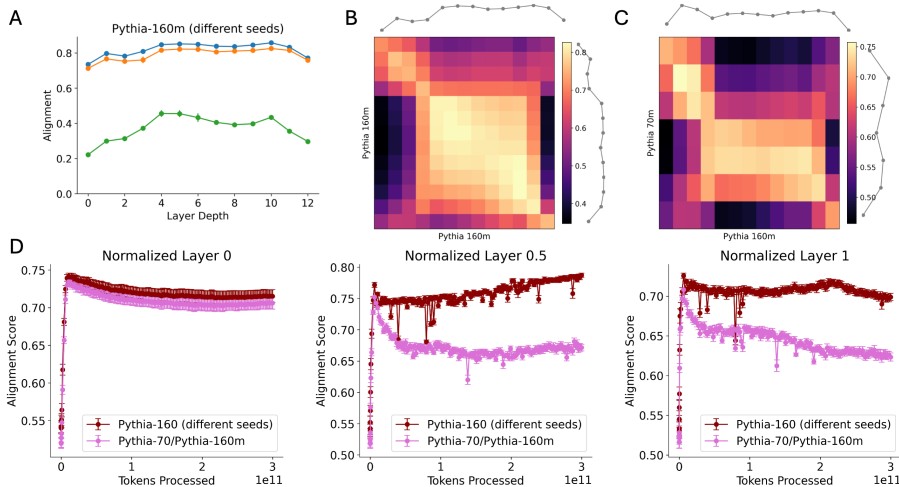

Figure 8: **Representational Convergence in Language Models. (A)** Alignment scores across all layers for Pythia-160m models trained from different seeds evaluated on STSB. Procrustes align representations nearly as well as linear transformations, mimicking the trend observed for vision models. **(B)** We compute the alignment scores using the Procrustes metric for every pair of layer in a seed-pair of Pythia-160m models. **(C)** Same as **(B)**, but for models with different architectures (varying depth). **(D)** Procrustes alignment between model pairs over training checkpoints.

**Rapid convergence:** alignment emerges early (layers at indices 0, 0.5, 1), peaks around 8–10B tokens, then plateaus or slightly declines (Fig. 8-D) even as next-token prediction improves up to 300B tokens [6].

# 5   Discussion

We present a large-scale account of convergent learning, showing how representational alignment between independently trained networks depends on depth, training time and distribution shifts. Using alignment metrics with varying degrees of transformation invariances for capturing representational similarity, we provide a more nuanced view of convergent learning.

Nonetheless, certain limitations remain. While we show the early emergence of alignment during training, we do not precisely quantify *when* within the first epoch this convergence occurs. Our empirical evidence for early alignment is striking and raises a crucial question: what drives such rapid convergence across diverse models? Although a full theoretical account is beyond our current scope, future work using tools like Neural Tangent Kernels (NTKs) [27], and its extension to CNNs [2]—under simplifying assumptions like linearity—may help elucidate the temporal dynamics of representational convergence. Another limitation stems from the alignment metrics themselves. Although our metrics reveal an early stabilization of alignment over training, this could reflect the limitations of the metrics rather than the absence of representational changes. Prior work [7] has shown that certain alignment metrics may fail to capture subtle, task-relevant shifts in representations. It remains possible that alternative metrics could reveal gradual alignment changes over training that are invisible to the methods employed here.

Finally, our method for computing alignment uses the center pixel of each feature map, enforcing a strict spatial correspondence between representations. This may underestimate alignment, especially in cases where features are slightly shifted spatially. While one could incorporate spatial shifts into alignment computations, they are computationally intensive and beyond the scope of this work. However, prior research has shown that optimal spatial shifts in many convolutional layers are typically negligible [46], justifying our approximation. Nonetheless, scalable methods that account for such spatial variability remain an important direction for future research.

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
