# OpenReview forum: "Bridging Critical Gaps in Convergent Learning: How Representational Alignment Evolves Across Layers, Training, and Distribution Shifts"
_NeurIPS.cc/2025/Conference — NeurIPS 2025 poster_

### Official Review · Reviewer_6h5w · 2025-06-30

**Clarity:** 4
**Significance:** 2
**Originality:** 2
**Rating:** 3
**Confidence:** 4

**Summary:**

This paper investigates convergent learning within neural networks. Here, inter-architectural similarity is evaluated across depth (layers) and convergence (different checkpoints across training), and metric (Procrustes, soft-matching, and linear regression).

They present a number of findings: 1) While the most flexible, linear regression shows only a minor increase in alignment in all situations (across depth / convergence) over orthogonal Procrustes, suggesting  2) Across most scenarios, representation similarity almost fully manifests within the first epoch of training 3) OOD behavior is observed mostly within the later layers of the network 4) Unlike CNN's, vision transformers do not seem to have a "privileged axis" where different instantiations of the same architecture converge to similar representations.

A few side experiments are run, with results such as: a) representation similarity between humans in the visual cortex decrease as information propagate through the visual pathway b) the similarity reported by soft-matching is often decreased when one network's activations are put through an orthogonal transformation, suggesting that actual axes along which information is computed and stored is retained across architectures c) much of their observations for supervised models are mirrored in self-supervised models.

**Questions:**

- Can you provide justification for taking the center values of the activation maps outside of computational efficiency? I'm quite concerned about how strong of claims can be made with so much of the representation thrown away.
- How does similarity change if an arbitrary spatial location is chosen for the activation, rather than the center?
- Would it be possible to evaluate similarity in a limited set of models / layers using the full activation map, compared to just the center channel, to understand the effect of that decision?
- I may be misrepresenting the novelty in my review, can you state what distinguishes this work from previous?

**Ethical Concerns:**

["NO or VERY MINOR ethics concerns only"]

**Final Justification:**

I'm maintaining my score of 3 as I'm still concerned about the use of the center activation in the feature map. The authors provided results taking random activations showing little change in results, though these were only for early layers (1-4) and not later ones--I think its reasonable to expect that the model is exploiting *less* spatial information early on (supported by papers I provided showing that use of spatial information builds in later layers).

Nonetheless, it should be taken into account that the other reviewers do not take issue in this. Ignoring this concern, I think the paper is fine.

**Limitations:**

yes

**Quality:**

2

**Strengths And Weaknesses:**

## Strengths
- No complaints about clarity and quality of writing. The paper was easy to follow throughout.
- In terms of experiments, the specific ones ran are reasonable ones to do to answer their questions of how representation similarity across architectures evolve with a) layer depth (Section 4.1) b) with training time (Section 4.2) c) distributions shifts (Section 4.3).
- I liked seeing evaluation of across layers, training time, and dataset all within one paper.
- Large-scale evaluation (across models and layers) provide broad empirical results.

## Weaknesses
- Choice of center activation: I think choosing the channels at the center of the networks representation might be a problem. I certainly understand the motivation, the activation maps of a CNN--especially early in the network--can be *extremely* large, computing similarity across thousands of layer-layer combinations might well be intractable without some way of reducing dimensionality. But in many cases it may be a massive reduction in the fidelity of the representation. As an example, I believe 224x224x64 is a reasonable early activation map for a ResNet50; if we take the center channel we retain about 0.002% of the data. From this tiny window into the network, strong, generalized claims about the convergent behavior of neural networks are made. The inductive biases of CNN's are all about translation equivariance, local feature detection, and hierarchical feature computation, but it seems like most of that would be discarded by taking the center channel. This may be reasonable choice early in the network layers, but does it hold higher up? [https://arxiv.org/abs/2110.14739](https://arxiv.org/abs/2110.14739) is cited as a possible justification for this, but my reading (e.g., Section 2.4) of this paper doesn't seem to support this conclusion. Further, this paper uses the entire activation map (though it is reshaped for computational convenience).
- The inductive biases of ViTs seem to encourage entirely distinct modes of computation compared to CNNs, it feels like an oversight to not include analysis across CNNs and ViTs.
- Novelty: Prior work has explored layer wise similarity and early training dynamics. Methodological novelty is limited, centering mostly on the three axis analysis.

My major concerns are the use of the center activation and novelty. If use of only the center activate is assumed to be an appropriate choice, I largely have no issues with the methodological approach and experimental results.

---

> ### Author Rebuttal · Authors · 2025-07-30
>
> We thank reviewer **6h5w** for the thoughtful comments and constructive suggestions. Below, we address each point in turn.
>
> > Choice of center activation
>
> We appreciate the reviewer’s concern and want to emphasize that comparing convolutional layers is nontrivial due to their spatial structure. Our choice of extracting activations from a single spatial location (the center pixel) was guided by both theoretical and practical considerations:
>
> 1. **Translation Equivariance of CNNs:** Convolutional layers are designed to detect the same features across all spatial locations via weight sharing. Each channel corresponds to a filter, and applying the filter across the image produces a spatial activation map. The full activation tensor is thus largely a *repeated sampling of the same feature dimensions at different locations*. With sufficient image samples, the distribution of feature responses at the center pixel is representative of the feature distribution across the entire map.
>
> 2. **Empirical Validation in Our Study:** In response to the reviewer’s suggestion, we have now repeated our analyses by sampling random spatial locations in each convolutional layer. The results closely match those obtained using the center pixel, confirming that this choice does not materially affect our findings. We will include these additional analyses in the Appendix of the revised manuscript.
>
> 3. **Theoretical Justification (ref. [1], Section 2.4):** As highlighted in **[1]**, the most rigorous way to compare convolutional layers would be to jointly optimize over both channel correspondences and spatial shifts between feature maps. By "spatial shifts," we mean that kernels detecting similar patterns can differ in their preferred spatial offset—for instance, one network’s $3\times 3$ kernel may respond to a $45^\circ$ edge centered in its receptive field, while another responds to the same edge slightly top-left. Fully aligning such features requires optimizing over both channels and spatial offsets, but this joint optimization is computationally intractable. Empirically, **[1]** shows that optimal spatial shifts are negligible in ConvNets trained on CIFAR10 (**Figure 3A** in Williams et al. (2019)), motivating our simpler choice to align activations only across channels using a fixed spatial position (center pixel) as a representative sample.
>
> 4. **Consistency with prior work:** Our strategy of sampling one spatial position per channel is consistent with earlier convergent‑learning studies. Li et al. **[4]** compute representational alignment by “**sampling a random image *and* a random position within the conv layer**” for each channel, i.e., they reduce every $h\times w\times c$ activation map to a single value per channel before alignment. We follow the same spirit—fixing the position to the center pixel for reproducibility and efficiency—while additionally verifying that random‑pixel sampling yields indistinguishable alignment trends.
>
> > The inductive biases of ViTs seem to encourage entirely distinct modes of computation compared to CNNs, it feels like an oversight to not include analysis across CNNs and ViTs
>
> We agree that understanding how vision transformers and CNNs converge (or fail to) is an important open question. In our work, we just discuss the presence/absence of privileged basis in CNNs vs transformers as a point of divergence. We focused on within‑architectural-family convergence because the methodological assumptions are clear. Extending the analysis across CNNs and ViTs raises non-trivial choices. ViTs operate on token sequences (often summarized by a $\texttt{[CLS]}$ token); CNNs yield spatial feature maps. There is no obvious one‑token analogue in CNNs, nor a principled way to up‑sample CNN features to match ViT tokens without injecting new invariances. Prior work (*e.g.,* [2]) appears to compare ViTs and CNNs using CKA **[3]** by flattening full feature maps for CNNs and using the $\texttt{[CLS]}$ token for ViTs (although they don’t spell this out exactly), but this implicitly weights spatial information very differently across architectures. Because a rigorous representational comparison protocol between ViTs and CNNs is still an open research problem and likely deserves a full study of its own, we restricted our claims to effects within CNNs and within ViTs. We now make this scope choice explicit in the manuscript and flag cross‑architectural-family alignment as a promising direction for future work.
>
> > I may be misrepresenting the novelty in my review, can you state what distinguishes this work from previous?
>
> Our work makes 3 novel contributions. First, no previous work has asked *what is the simplest mapping that still aligns two networks?* We answer this by applying three metrics with progressively weaker invariances (permutation $\rightarrow$  orthogonal $\rightarrow$  linear) and find that orthogonal transformations—which preserve geometric structure—align networks nearly as effectively as linear mappings, suggesting simple rotations/reflections drive convergence, not complex affine distortions. This exposes a far tighter structural similarity than earlier CCA‑ or regression‑based studies. Second, we document a surprising temporal dynamic: convergence largely occurs in the first epoch, challenging the view that alignment arises solely from prolonged task optimization. Prior work on “early‑phase” dynamics examined single networks; no study has shown such *cross‑network* alignment emerging this early, nor replicated it across multiple models and domains (vision/language). Third, by systematically examining the effects of distribution shifts, we find that early layers robustly maintain alignment while later layers diverge significantly under OOD conditions in direct proportion to the distribution shift. To our knowledge, these findings have not been reported before. These findings come from a large‑scale audit  that (i) spans CNNs, ViTs, and LLMs; (ii) tests three alignment families (permutation, Procrustes, linear regression) under identical protocols; and (iii) reports thousands of layer‑pair comparisons—far broader than earlier studies, which typically used a single metric, a handful of models, and a single checkpoint.
>
> **References:**
>
> **[1]** Generalized Shape Metrics on Neural Representations (Williams et al., 2019)
>
> **[2]** Do Vision Transformers See Like Convolutional Neural Networks (Raghu et al., 2022)
>
> **[3]** Similarity of Neural Network Representations Revisited (Kornblith et al., 2019)
>
> **[4]** Convergent Learning: Do Different Neural Networks Learn the Same Representations (Li et al., 2015)

---

> ### Comment · Reviewer_6h5w · 2025-08-02
>
> Thanks for the response, you've suitably addressed my concerns w.r.t ViTs and novelty, though I still have some reservations about the center activation choice. CNN's are built on translation invariant operation, yet they seem to exploit and encode spatial information [1, 2]. I'm not sure if I can agree with the statement that "With sufficient image samples, the distribution of feature responses at the center pixel is representative of the feature distribution across the entire map." Nonetheless, if sampling random positions does in fact not materially change the results then I'm happy to walk this concern back as well. Though I would like to see results you mentioned first.
>
>
> [1] Kayhan, Osman Semih and Jan C. van Gemert. “On Translation Invariance in CNNs: Convolutional Layers Can Exploit Absolute Spatial Location.” 2020 IEEE/CVF Conference on Computer Vision and Pattern Recognition (CVPR) (2020): 14262-14273.
>
> [2] Islam, Md. Amirul et al. “How Much Position Information Do Convolutional Neural Networks Encode?” ArXiv abs/2001.08248 (2020): n. pag.

---

> > ### Author Response · Authors · 2025-08-04
> >
> > We thank reviewer **6h5w** for acknowledging that our response regarding ViTs and novelty addresses their respective concerns. Because OpenReview does not permit embedding figures in rebuttals, we have instead tabulated the alignment scores for our key results under both sampling schemes: the original center-pixel choice and a random pixel. For brevity, we show the results from a single model-seed pair (ResNet18) but as noted in our previous response, we will include the complete set of results in our revised manuscript. The numbers shift only marginally across the two sampling methods, and every qualitative conclusion persists: **(i)** alignment still declines with depth, **(ii)** Procrustes continues to close nearly the full gap to Linear, **(iii)** almost all of the final alignment appears in the first epoch, and **(iv)** distribution shifts affect alignment of later but not early layers. These results confirm that pixel choice does not confound any of the paper’s main findings.
> >
> >
> > ### Layerwise model alignment scores (center vs. random pixels):
> > | Layer Name       | Center (Linear / Ortho / Perm) | Random (Linear / Ortho / Perm) |
> > |:------------------:|:-------------------------------:|:--------------------------------:|
> > | $\texttt{conv1}$            | $0.996 / 0.967 / 0.721$          | $0.998 / 0.968 / 0.724$          |
> > | $\texttt{layer1.0.conv1}$   | $0.963 / 0.930 / 0.694$          | $0.964 / 0.932 / 0.696$          |
> > | $\texttt{layer1.0.conv2}$   | $0.904 / 0.844 / 0.572$          | $0.904 / 0.846 / 0.574$          |
> > | $\texttt{layer1.1.conv1}$   | $0.900 / 0.842 / 0.569$          | $0.909 / 0.854 / 0.581$          |
> > | $\texttt{layer1.1.conv2}$   | $0.772 / 0.695 / 0.422$          | $0.780 / 0.704 / 0.432$          |
> > | $\texttt{layer2.0.conv1}$   | $0.879 / 0.827 / 0.524$          | $0.878 / 0.826 / 0.525$          |
> > | $\texttt{layer2.0.conv2}$   | $0.788 / 0.726 / 0.441$          | $0.848 / 0.778 / 0.460$          |
> > | $\texttt{layer2.1.conv1}$   | $0.819 / 0.761 / 0.470$          | $0.816 / 0.758 / 0.471$          |
> > | $\texttt{layer2.1.conv2}$   | $0.712 / 0.634 / 0.349$          | $0.707 / 0.630 / 0.350$          |
> > | $\texttt{layer3.0.conv1}$   | $0.825 / 0.770 / 0.424$          | $0.823 / 0.771 / 0.427$          |
> > | $\texttt{layer3.0.conv2}$   | $0.759 / 0.692 / 0.355$          | $0.747 / 0.681 / 0.351$          |
> > | $\texttt{layer3.1.conv1}$   | $0.772 / 0.707 / 0.369$          | $0.757 / 0.691 / 0.366$          |
> > | $\texttt{layer3.1.conv2}$   | $0.711 / 0.632 / 0.335$          | $0.718 / 0.640 / 0.348$          |
> > | $\texttt{layer4.0.conv1}$   | $0.740 / 0.671 / 0.352$          | $0.726 / 0.654 / 0.338$          |
> > | $\texttt{layer4.0.conv2}$   | $0.721 / 0.645 / 0.268$          | $0.691 / 0.619 / 0.260$          |
> > | $\texttt{layer4.1.conv1}$   | $0.771 / 0.698 / 0.465$          | $0.696 / 0.613 / 0.335$          |
> > | $\texttt{layer4.1.conv2}$   | $0.653 / 0.574 / 0.283$          | $0.625 / 0.564 / 0.294$          |
> >
> >
> > ### (Mean) OOD Alignment for center vs. random pixel (OOD stimulus set):
> >
> > |     Layer Name     |       Center (Ortho)        |       Random (Ortho)        |
> > |:------------------:|:---------------------------:|:----------------------------:|
> > | $\texttt{conv1}$              | $0.954$       | $0.967$    |
> > | $\texttt{layer1.0.conv1}$     | $0.911$       | $0.919$    |
> > | $\texttt{layer1.0.conv2}$     | $0.843$       | $0.854$    |
> > | $\texttt{layer1.1.conv1}$     | $0.829$       | $0.836$    |
> > | $\texttt{layer1.1.conv2}$     | $0.704$       | $0.720$    |
> > | $\texttt{layer2.0.conv1}$     | $0.797$       | $0.799$    |
> > | $\texttt{layer2.0.conv2}$     | $0.686$       | $0.708$    |
> > | $\texttt{layer2.1.conv1}$     | $0.725$       | $0.742$    |
> > | $\texttt{layer2.1.conv2}$     | $0.589$       | $0.620$    |
> > | $\texttt{layer3.0.conv1}$     | $0.714$       | $0.734$    |
> > | $\texttt{layer3.0.conv2}$     | $0.607$       | $0.634$    |
> > | $\texttt{layer3.1.conv1}$     | $0.627$       | $0.654$    |
> > | $\texttt{layer3.1.conv2}$     | $0.515$       | $0.542$    |
> > | $\texttt{layer4.0.conv1}$     | $0.562$       | $0.290$    |
> > | $\texttt{layer4.0.conv2}$     | $0.510$       | $0.498$    |
> > | $\texttt{layer4.1.conv1}$     | $0.597$       | $0.555$    |
> > | $\texttt{layer4.1.conv2}$     | $0.466$       | $0.440$    |

---

> > > ### Author Response · Authors · 2025-08-04
> > >
> > > (continued response)
> > >
> > > ### Alignment as a function of training epochs
> > >
> > > | Epoch | Layer Name          | Center (Ortho) | Random (Ortho) |
> > > |:-----:|:--------------------|:--------------:|:--------------:|
> > > | $0-0$ | $\texttt{conv1}$ (Normalized layer: $0$)     |        $0.834$          | $0.835$          |
> > > |       | $\texttt{layer2.1.conv2}$ (Normalized layer: $0.5$)| $0.295$          | $0.383$          |
> > > |       | $\texttt{layer4.1.conv2}$ (Normalized layer: $1$)  | $0.152$          | $0.169$          |
> > > | $1-1$ | $\texttt{conv1         }$      | $0.915$          | $0.918$          |
> > > |       | $\texttt{layer2.1.conv2}$      | $0.685$          | $0.687$          |
> > > |       | $\texttt{layer4.1.conv2}$      | $0.697$          | $0.698$          |
> > > | $2-2$ | $\texttt{conv1         }$      | $0.922$          | $0.934$          |
> > > |       | $\texttt{layer2.1.conv2}$      | $0.687$          | $0.686$          |
> > > |       | $\texttt{layer4.1.conv2}$      | $0.684$          | $0.681$          |
> > > | $3-3$ | $\texttt{conv1         }$      | $0.929$          | $0.932$          |
> > > |       | $\texttt{layer2.1.conv2}$      | $0.666$          | $0.717$          |
> > > |       | $\texttt{layer4.1.conv2}$      | $0.657$          | $0.657$          |
> > > | $4-4$ | $\texttt{conv1         }$      | $0.931$          | $0.932$          |
> > > |       | $\texttt{layer2.1.conv2}$      | $0.666$          | $0.678$          |
> > > |       | $\texttt{layer4.1.conv2}$      | $0.652$          | $0.654$          |
> > > | $5-5$ | $\texttt{conv1         }$      | $0.933$          | $0.936$          |
> > > |       | $\texttt{layer2.1.conv2}$      | $0.662$          | $0.676$          |
> > > |       | $\texttt{layer4.1.conv2}$      | $0.627$          | $0.622$          |
> > > | $6-6$ | $\texttt{conv1         }$      | $0.933$          | $0.935$          |
> > > |       | $\texttt{layer2.1.conv2}$      | $0.657$          | $0.669$          |
> > > |       | $\texttt{layer4.1.conv2}$      | $0.624$          | $0.617$          |
> > > | $7-7$ | $\texttt{conv1         }$      | $0.933$          | $0.936$          |
> > > |       | $\texttt{layer2.1.conv2}$      | $0.657$          | $0.669$          |
> > > |       | $\texttt{layer4.1.conv2}$      | $0.630$          | $0.623$          |
> > > | $8-8$ | $\texttt{conv1         }$      | $0.931$          | $0.935$          |
> > > |       | $\texttt{layer2.1.conv2}$      | $0.644$          | $0.645$          |
> > > |       | $\texttt{layer4.1.conv2}$      | $0.621$          | $0.616$          |
> > > | $9-9$ | $\texttt{conv1         }$      | $0.936$          | $0.937$          |
> > > |       | $\texttt{layer2.1.conv2}$      | $0.645$          | $0.656$          |
> > > |       | $\texttt{layer4.1.conv2}$      | $0.620$          | $0.617$          |

---

### Official Review · Reviewer_CU7z · 2025-06-30

**Clarity:** 2
**Significance:** 3
**Originality:** 3
**Rating:** 5
**Confidence:** 4

**Summary:**

This paper provides a large-scale audit of convergence learning, focusing on three alignment metrics and disentangling the influence of layer position, training phase, and distribution shifts. The main paper provides results for convolutional networks, whereas more networks (self-supervised vision models, vision transformers, and language models) are deferred to the appendix.

**Questions:**

1) What are the key findings that are shared across all model types? Are there any big differences?
2) Can you address some of the questions regarding section 4.3 (see the weaknesses).
3) In Table 1, why do you report min and max alignment scores, and not, for instance, mean and standard deviation? Do you think it better summarizes the results? To me, it feels like I am missing some important information to fully understand what the results look like.

**Ethical Concerns:**

["NO or VERY MINOR ethics concerns only"]

**Final Justification:**

It is hard to assess the readability and organization of the paper, updated as the authors promised. However, the provided comments and excerpt from the discussion make me believe that these changes will improve the paper and highlight the main contributions of the paper.
The authors addressed all my questions. I think the paper is a nice contribution, tackling an important and timely problem. I do not raise the score because I do not think the impact of the paper will be groundbreaking, but I think it is a solid paper and support its acceptance.

**Limitations:**

Yes

**Paper Formatting Concerns:**

No issues

**Quality:**

3

**Strengths And Weaknesses:**

**Strengths:**
- The motivation for the paper is clear, and the problem it investigates is important. Thorough studies comparing multiple variables and not just one method evaluated on one dataset can provide deeper insights, objectively compare the methods, and isolate the key factors.
- The methods are clearly described, and the number of experiments conducted (including the ones in the appendix) is sufficient. All claims and conclusions are supported by the provided figures and tables.
- The results are put into the context of relevant literature, the authors comment on them well, and provide possible explanations.

**Weaknesses:**
- I think the biggest weakness is how the results are organized and presented.
	- For instance, the main paper contains results only for convolutional models, whereas self-supervised vision and language models are deferred to the appendix. I think the paper could benefit from reorganizing. The key results and conclusions holding across model types could be in the main paper, and the detailed results could be moved to the appendix.
	- It is a bit hard to identify the key findings (partially because there is no conclusion section). I found many interesting findings and claims in the results section, but it seems to me that they are not highlighted enough in the other sections, so a reader might easily overlook them unless they read the whole paper carefully.
- Figures should contain all the information needed to understand them (for example, what each row and column means). It is described in the captions, but one has to read the whole paragraph to find all the information, for instance, the data it was evaluated on.
-  I struggle to fully understand section 4.3. Some of my questions might just need a clearer explanation, and some are challenging the whole experimental design.
	- If I understand it correctly, you use only the Procrustes metric in this part. Why not all three of them?
	- In Figure 5, do you compare only the same layers (at the same depth)? Why not all of them? Could it be the case that the networks process the OOD data slightly differently from the WD data, so some of the processing might actually happen at different layers? Also, just from the figure, the difference does not seem to be that big. Did you test the significance of the results?
	- I do not understand Figure 6. In a), what is the task performance? Is it layer-specific, or (as I guess) one number for the whole network? It is slightly confusing to plot one information twice. How exactly is b) computed?
	- You claim that "... one could plausibly improve OOD generalisation by fine-tuning only the later layers." and "...OOD stimuli could be especially useful to distinguish and select between models whose representations closely mirror the brain." Could you provide more details about this? Have you (or some other work) investigated that?

---

> ### Author Rebuttal · Authors · 2025-07-30
>
> We would sincerely like to thank reviewer **Cu7z** for their thoughtful comments and recognizing the importance of our work. Below, we address each of their concerns:
>
> > I think the biggest weakness is how the results are organized and presented.
>
> We agree with the reviewer that the results could be reorganized in the paper to highlight our main contributions. We have now reorganized the manuscript to include all CIFAR-100 results in the appendix instead of the main text. We have also moved our results on self-supervised networks, Vision Transformers and language models to the main text. We also reiterate our novel contributions in the **Discussion section (Sec. 5)** and make our contributions verbose. Specifically, we highlight that our work makes 3 novel contributions. First, no previous work has asked *what is the simplest mapping that still aligns two networks?* We answer this by applying three metrics with progressively weaker invariances (permutation $\rightarrow$ orthogonal $\rightarrow$ linear) and find that orthogonal transformations—which preserve geometric structure—align networks nearly as effectively as linear mappings, suggesting simple rotations/reflections drive convergence, not complex affine distortions. This exposes a far tighter structural similarity than earlier CCA‑ or regression‑‑based studies. Second, we document a surprising temporal dynamic: convergence largely occurs in the first epoch, challenging the view that alignment arises solely from prolonged task optimization. Prior work on “early‑phase” dynamics examined *single* networks; no study has shown such *cross‑network* alignment emerging this early, nor replicated it across multiple models and domains (vision/language). Third, by systematically examining the effects of distribution shifts, we find that early layers robustly maintain alignment while later layers diverge significantly under OOD conditions in direct proportion to the distribution shift. To our knowledge, these findings have not been reported before. These findings come from a large‑scale audit  that **(i)** spans CNNs, ViTs, and LLMs; **(ii)** tests three alignment families (permutation, Procrustes, linear regression) under identical protocols; and **(iii)** reports thousands of layer‑pair comparisons—far broader than earlier studies, which typically used a single metric, a handful of models, and a single checkpoint.
>
> > Figures should contain all the information needed to understand them
>
> Thank you for pointing this out, we have now added legends to subplots as well as titles such that information about the experimental results need not be necessarily read from their respective captions.
>
> > If I understand it correctly, you use only the Procrustes metric in this part. Why not all three of them?
>
> Indeed, all three alignment metrics—linear regression, orthogonal Procrustes, and permutation/soft-matching—show consistent trends throughout our analyses. We chose to use the Procrustes metric in **Figures 5** and **6** as it offers a principled trade-off between mapping flexibility and interpretability: it discounts only rotations and reflections, and as shown in our earlier results, it achieves alignment scores that are nearly as high as linear regression, indicating that most differences across networks can be accounted for by simple rigid transformations. We have clarified this rationale in the manuscript and included the results for linear predictivity and permutation/soft-matching in the appendix.
>
> > In Figure 5, do you compare only the same layers (at the same depth)? Why not all of them? Could it be the case that the networks process the OOD data slightly differently from the WD data, so some of the processing might actually happen at different layers? Also, just from the figure, the difference does not seem to be that big. Did you test the significance of the results?
>
> We wanted to maintain consistency with our remaining analyses, by comparing layers at the same depth across two independently trained models. To address the reviewer’s point, we have now computed full layer-by-layer alignment matrices for every OOD dataset. In all cases the strongest alignment still lies on the diagonal, indicating that OOD inputs do not shift processing to different depths; the hierarchical correspondence mirrors that observed with in‑distribution data.
>
> Further, for significance, we now performed a relative $t$-test comparing WD vs. OOD alignment separately on the “early” layers (normalized depth $< 0.5$) and “late” layers (normalized depth $> 0.5$) across all architectures. The average $p$-values were $0.038$ for early layers and $0.0006$ for late layers, confirming that deeper layers diverge significantly more under distribution shifts. We will include the significance table in the revised manuscript.
>
>
> > I do not understand Figure 6. In a), what is the task performance? Is it layer-specific, or (as I guess) one number for the whole network? It is slightly confusing to plot one information twice. How exactly is b) computed?
>
> In **Fig. 6a**, each value in the scatter plot represents the overall task performance of the full network on a given OOD dataset **[1]**. Individual points correspond to different OOD datasets, as noted in the figure legend, and each subplot reflects a specific ResNet50-layer indicated in the subplot title. The $y$-axis depicts the Procrustes alignment score between two independently initialized ResNet‑50s at the layer named in the subplot title (*e.g.,* “conv1” or “penultimate”). Our goal with this figure is to demonstrate that early layers stay highly aligned regardless of OOD accuracy, whereas deeper layers’ alignment covaries strongly with the accuracy drop.
>
> For **Fig. 6b**, we first normalize the layer depths to a common scale $(0-1)$ across all architectures (ResNet18, ResNet50, VGG16 and VGG19) so that their line plots can be directly compared. We then compute for each layer $\ell$ the correlation $\rho(\ell)$ across the $17$ datasets between **(i)** network-level OOD accuracy (dataset) and **(ii)** alignment score ($\ell$, dataset) as a function of normalized layer depth across each model's entire depth. This analysis demonstrates how the coupling between representational divergence and performance grows along the hierarchy
>
> > You claim that "... one could plausibly improve OOD generalisation by fine-tuning only the later layers." and "...OOD stimuli could be especially useful to distinguish and select between models whose representations closely mirror the brain." Could you provide more details about this? Have you (or some other work) investigated that?
>
> We apologise for any confusion—those remarks were meant as forward‑looking hypotheses rather than firm conclusions. The direct evidence for "fine-tuning only later layers improves OOD generalization" appears to be limited. We inferred the idea from the observations that **(i)** early CNN layers capture generic, transferable features whereas **(ii)** deeper layers become task‑specific, so preserving the former while adapting the latter should in principle help under distribution shift. Likewise, our suggestion that OOD stimuli could better differentiate models was also speculative. Isolated findings already hint at this leverage: Kar et al. (2019) showed that recurrent and feed‑forward CNNs, indistinguishable on simple images, separate when tested on challenging scenes; To our knowledge, however, no work has systematically compared “natural” versus distorted image sets for the explicit purpose of ranking brain‑predictive models.
>
> > What are the key findings that are shared across all model types? Are there any big differences?
>
> Across every model family we tested, three patterns were universal: **(i)** convergence emerges almost immediately: most of the eventual alignment is in place after the first training epoch; **(ii)** orthogonal maps are nearly as good as full linear maps, indicating that simple rotations/reflections account for most representational differences; and **(iii)** early layers are both highly aligned and robust to distribution shift, whereas deeper layers diverge in proportion to OOD accuracy drops. The first two results also extended to the language domain. The main divergences appear along architectural lines: in vision models alignment falls monotonically with depth, but in language models this trend is not apparent; CNNs exhibit clear “privileged axes” (permutation alignment well above chance), whereas ViTs do not show axis preservation.
>
> > In Table 1, why do you report min and max alignment scores, and not, for instance, mean and standard deviation? Do you think it better summarizes the results? To me, it feels like I am missing some important information to fully understand what the results look like.
>
> We chose min / max values in **Table 1** because we wanted to know whether *every* layer shows privileged‑axis behavior. Reporting the **minimum** alignment after rotation tells us whether even the *least* aligned layer still exhibits a substantial drop (i.e., no layer is exempt), while the **maximum** establishes how large the effect can be. A mean $\pm$  SD would blur this worst‑case information; a single high‑variance layer could be “averaged away.” That said, we agree the reader may also want a sense of central tendency, so in the revision we now provide mean $\pm$ SD across layers in the Appendix and note that the qualitative conclusion—privileged axes at every layer—remains unchanged.
>
> **References:**
>
> **[1]** Partial success in closing the gap between human and machine vision (Geirhos et al., 2021)
>
> **[2]** Evidence that recurrent circuits are critical to the ventral stream's execution of core object recognition behavior (Kar et al., 2019)

---

> > ### Comment · Reviewer_CU7z · 2025-08-01
> >
> > I thank the authors for addressing my comments. It is hard to estimate the impact of the changes on the structure and readability of the paper without seeing the updated version, but I hope it will be a great improvement.

---

### Official Review · Reviewer_uReV · 2025-07-03

**Clarity:** 2
**Significance:** 2
**Originality:** 2
**Rating:** 4
**Confidence:** 3

**Summary:**

This paper aims to understand convergent learning and proposes a large-scale audit for it.  The authors conduct an extensive experiment, which demonstrates the usefulness of the proposed method.

**Questions:**

1. What is the specific expression of the $\texttt{corr}$ formula in Line 134?

2. The symbol $\mathcal{O}$ is used repeatedly in Line 163, with different meanings in different places (Line 129).

3. The method part seems to lack a lot of core formula expressions. Should you migrate them from the problem statement?

4. The legend font in all the figures is too small.

5. Is there enough theory to support the effectiveness of the proposed method? Specific analysis is needed.

**Ethical Concerns:**

["NO or VERY MINOR ethics concerns only"]

**Final Justification:**

After reading the author's response to us and other reviewers, we are confident that the author has addressed most of our concerns appropriately, so we decided to increase our final score.

**Limitations:**

yes

**Quality:**

2

**Strengths And Weaknesses:**

**Strengths**

1. This paper proposes a comparative method of characterization from three perspectives: layer, training, and distribution shift.

2. Real data analysis demonstrated the impact of the new methods.

**Weaknesses**

1. The main concern of the paper is a presentation. In many places, the meaning of sentences is vague. It is better to elaborate on it and make the sentence shorter.

2. The paper involves numerous symbols and matrix operations, but some symbols lack clear explanations. It is recommended to define each symbol's meaning the first time it appears to avoid ambiguity.

3. All formulas in the article lack numbers, which makes it very inconvenient to quote and explain.

4. The motivation for the proposed method is not well explained, and the novelty of the method seems to be underdeveloped.

---

> ### Author Rebuttal · Authors · 2025-07-30
>
> We would like to thank reviewer **uRev** for taking their time to review our manuscript. Below, we have addressed each point in turn, and tried to clarify some major misunderstandings:
>
> **General Remark**
>
> First, we respectfully clarify that the novelty of our work lies in its **scale and scope of empirical inquiry**, not in proposing a new algorithm. Specifically, we deliver the first systematic audit of representational convergence across three orthogonal dimensions—layer depth, training trajectory, and distribution shift—spanning CNNs, ViTs, and LLMs.
>
> **Response to Weaknesses**:
>
> 1. > The main concern of the paper is a presentation. In many places, the meaning of sentences is vague. It is better to elaborate on it and make the sentence shorter.
>
> We appreciate the reviewers' concern to help improve our manuscript presentation. Because the comment does not specify which passages are unclear, we are unable to address them individually in the rebuttal. We will, however, undertake a careful language edit for the camera‑ready version and would welcome any concrete pointers the reviewer can provide.
>
> 2. > The paper involves numerous symbols and matrix operations, but some symbols lack clear explanations. It is recommended to define each symbol's meaning the first time it appears to avoid ambiguity.
>
> Our manuscript describes mathematical equations in Section 2. We believe all symbols are already defined at first mention in Section 2. the only exception is the transportation polytope $\mathcal{T}(N_x, N_y)$, for which we cited prior work. In the revision we will **(i)** write out the explicit linear constraints that define $\mathcal{T}(N_x, N_y)$ and **(ii)** add a one‑page notation table in the appendix for quick reference. If any other expression remains unclear, we would welcome a specific pointer so we can correct it.
>
> 3. > All formulas in the article lack numbers, which makes it very inconvenient to quote and explain.
>
> We have omitted the numbering of equations because no equation was referenced more than once. However, if the reviewer feels that it is necessary to do so for clarity, we can make these changes.
>
> 4. Our manuscript does not introduce a new alignment algorithm; instead, it offers a systematic experimental framework to interrogate long‑standing questions about representational convergence. The motivation is stated up front: prior studies typically (i) examine only one metric, (ii) snapshot a single checkpoint, and (iii) ignore distribution shifts—leaving open **(i)** what minimal transformations are sufficient to align representations, **(ii)** when convergence first appears during training, and **(iii)** how it behaves under distribution shift.. We fill these gaps by applying three established metrics side‑by‑side across thousands of layer‑pair comparisons, tracking the entire training trajectory, and evaluating $17$ OOD datasets. This scale and breadth reveal previously unreported phenomena—*e.g.,* rotation suffices, convergence appears in the first epoch, early layers remain aligned under OOD shift—thereby providing new insights rather than a new method.
>
> Next, we address each of the reviewers questions below:
>
> 1. $ \texttt{Alignment} = \texttt{corr}(\boldsymbol{X}_j, \boldsymbol{MX}_i)$ is used to report representational alignment as pearson correlation computed unit-wise across stimuli. After applying the optimal mapping $\boldsymbol{M}$ to $\boldsymbol{X}_i$​, we take each corresponding pair of columns—$\boldsymbol{x}_j(k)$ from $\boldsymbol{X}_j$ and $\boldsymbol{Mx}_i(k)$ from $\boldsymbol{MX}_i$ (where each column is a length‑$\boldsymbol{M}$ response vector over the same set of stimuli)—and compute the Pearson correlation coefficient $r_k = \texttt{corr⁡} ⁣(\boldsymbol{x}_j(k), \boldsymbol{ Mx}_i(k))$ where $\texttt{corr}(a, b) = \frac{\sum{(a_i - \bar{a})(b_i - \bar{b})}}{\sqrt{\sum{(a_i - \bar{a})^2} \sum{(b_i - \bar{b})^2}}}$. The reported alignment score is the mean of these per‑unit correlations: $\texttt{Alignment}(\boldsymbol{X}_i, \boldsymbol{X}_j) = \frac{1}{\min{(N_x, N_y)}}\sum r_k$. We have inserted this explicit definition in the revised manuscript.
>
> 2. The symbol $\mathcal{O}$ is used in **L129** as standard notation to describe the standard $n$ dimensional orthogonal group. However, in **L163**, $\mathcal{O}$ is yet again standard notation to describe computational time complexity (or Big-O notation). To avoid any ambiguity we will use **two distinct fonts: the bold‑face roman capital “$\mathbf{O}$” for the orthogonal group and the calligraphic “$\mathcal{O}$” for big-O notation.**
>
> 3. Thank you for pointing this out, we will make the text in the legends larger in the revised version of our manuscript.
>
> 4. Our study does **not** propose a new algorithmic method that would require novel theoretical guarantees; rather, it is an empirical audit that employs three well‑established alignment metrics (linear regression, orthogonal Procrustes, permutation/soft‑matching). The theoretical properties of these metrics—*e.g.,* invariance classes, optimization uniqueness—are already documented in the literature we cite (Williams et al., 2019; Khosla & Williams,  2024). Our contribution is to apply them side‑by‑side at unprecedented scale and reveal new empirical regularities (minimal gap between procrustes and linear predictivity, first‑epoch convergence, OOD divergence for later layers). Therefore, additional formal analysis is not needed to support the effectiveness of the metrics themselves; the value of our work lies in the comprehensive evidence we present.
>
> **References:**
>
> **[1]** Generalized Shape Metrics on Neural Representations (Williams et al., 2019)
>
> **[2]** Soft Matching Distance: A metric on neural representations that captures single-neuron tuning (Khosla & Williams, 2024)

---

> > ### Comment · Reviewer_uReV · 2025-08-05
> >
> > Thank you for your detailed reply. Having read your responses to the other reviewers and the substantially expanded revised manuscript, I think this paper should be accepted, and will therefore raise my score.

---

### Official Review · Reviewer_yuMg · 2025-07-03

**Clarity:** 3
**Significance:** 3
**Originality:** 2
**Rating:** 5
**Confidence:** 4

**Summary:**

This work seeks to understand how representations in various neural networks converge according to various metrics across datasets, architectures, throughout training, and in out of distribution settings. Through a large scale study, authors reveal that 1) representational convergence occurs even if the mapping function is quite restricted 2) convergence occurs extremely early in training, 3) that OOD stimuli disrupt representational convergence in later layers and 4) convergence occurs even between different CNN architectures.

**Questions:**

Add Spearman correlation to sanity check results.

Add randomly initialized baseline models.

What produces basis alignment in CNNs as opposed to transformers?

**Ethical Concerns:**

["NO or VERY MINOR ethics concerns only"]

**Final Justification:**

The authors ran the relevant extra analyses and baselines during the rebuttal period (random model and Spearman correlation), addressing my major technical concerns with the paper.

**Limitations:**

Yes.

**Quality:**

3

**Strengths And Weaknesses:**

This study is quite extensive and thoughtful, revealing several notable findings that challenge prevailing wisdom regarding convergence. Most notable among them is that representations tend to converge within the first epoch of training. That said, there are a few opportunities to improve the work, and some sanity checks that might be needed to ensure their validity.

First, it seems plausible that the correlation between representations could be driven by the presence of rogue/outlier dimensions (https://arxiv.org/abs/2310.17715). This could especially be the case with the permutation-based alignment score. Outlier dimensions are thought to encode generic task information in extremely high variance dimensions, and so it is possible that such dimensions always encode approximately the same basic information across datasets/architectures/etc. As I understand it, the correlation score is presently simply Pearson correlation, which is susceptible to such outliers. A simple sanity check is to replace Pearson correlation with Spearman correlation, and ensure that all results still hold.
Additionally, the results from Figure 1 should include a baseline randomly initialized model to contextualize the observed trends.

Writing:
The Difference (%) column in table 1 could be more explicitly described in text.

The neuroscience framing seems largely unnecessary in this paper, and could very well be distracting to readers unfamiliar with current paradigms in that field. While the compneuro community might benefit from this work, it is unclear that framing paper around the neuroscientific applications is more compelling than immediately (i.e., in the intro) putting this work in conversation with the debates surrounding the platonic representation hypothesis in AI.

---

> ### Author Rebuttal · Authors · 2025-07-30
>
> We would like to thank reviewer **yuMg** for recognizing the novelty of our results! It is indeed quite remarkable that representational convergence emerges within the first epoch itself.
>
> > As I understand it, the correlation score is presently simply Pearson correlation, which is susceptible to such outliers. A simple sanity check is to replace Pearson correlation with Spearman correlation, and ensure that all results still hold.
>
> We appreciate the reviewers' concern about Pearson correlation. While our use of Pearson correlation follows prior work in representational alignment, we agree that it is susceptible to high-variance outlier dimensions. To directly address this possibility, we conducted an additional series of experiments where we computed alignment scores using Spearman’s rank correlation instead of Pearson’s correlation. Across all models and metrics, we observed that the alignment trends over network depth remained qualitatively unchanged, and all key findings about early emergence of convergence and effect of distribution shifts still held. These results indicate that our conclusions are not driven by a small number of high-variance dimensions. We will include the corresponding figures in the revised supplementary material for completeness.
>
> >  Additionally, the results from Figure 1 should include a baseline randomly initialized model to contextualize the observed trends.
>
>
> We appreciate this suggestion and have now extended our analysis to include randomly initialized (untrained) networks as a baseline. We find that alignment scores across all metrics are consistently lower in untrained networks compared to their trained counterparts. This difference is especially pronounced in deeper layers: for example, using the Procrustes metric, the mean alignment increases by $145.26\\%$ in early layers (depth $< 0.5$) and by $493.84\\%$ in deeper layers (depth $> 0.5$) following training.
>
> Additionally, we observe that in untrained networks, the mean gap between linear and Procrustes alignment is substantially larger $(42.39\\% )$ than that seen in trained models $(7.85\\%)$. This suggests that training progressively aligns representations in a way that can be well captured by rigid transformations (i.e., rotations/reflections), reducing the need for more flexible affine transformations. Together, these findings reinforce the conclusion that representational convergence is not simply a consequence of initialization biases or mere architectural biases, but is shaped through training. We will include these results in the revised supplementary materials.
>
> > The neuroscience framing seems largely unnecessary in this paper, and could very well be distracting to readers unfamiliar with current paradigms in that field
>
> We appreciate the reviewer’s perspective. Our original motivation for including the neuroscience framing was to highlight the broader relevance of representational convergence—not only as a phenomenon observed across artificial neural networks, but also as a question of interest in systems neuroscience, where similar analyses are used to compare brain responses across individuals and species. However, we agree that the neuroscience context is not essential to the core contributions of this work, and may distract readers focused on machine learning. In response, we have revised the manuscript to remove neuroscience-specific framing and now focus exclusively on representational convergence between models.
>
> > Writing: The Difference (%) column in table 1 could be more explicitly described in text.
>
> We have revised the corresponding paragraph to more clearly describe the “Difference $(\\%)$” column in **Table 1**. Specifically, we now clarify that this column reports the relative decrease in permutation alignment scores after applying a random rotation to the representational basis, computed as the percentage difference between native and rotated alignment scores for each layer.
>
> > What produces basis alignment in CNNs as opposed to transformers?
>
> While a definitive mechanistic account remains an open area of investigation, recent work **[1]** offers compelling evidence that the emergence of basis alignment across CNNs and even between brains and CNNs—may be partly attributable to architectural choices, especially the presence of ReLU nonlinearities. To understand this constraint, consider a representation of post-ReLU activations $\boldsymbol{x}$. The ReLU operation ensures that all activations in $\boldsymbol{x}$ are non-negative. If we apply a rotation to these activations, we get $\boldsymbol{y} = \boldsymbol{Ux}$, where $\boldsymbol{U}$ is a rotation matrix. For both $\boldsymbol{x}$ and $\boldsymbol{y}$ to be valid post-ReLU activations, they must remain non-negative after the transformation. For this, the matrix $\boldsymbol{U}$ must be a non-negative matrix. But this means that $\boldsymbol{U}$ must be a permutation matrix, because every orthogonal matrix with nonnegative entries is necessarily a permutation matrix. This means that $\boldsymbol{U}$ can only permute (or shuffle) the activation units, rather than performing arbitrary rotations. Thus, the nonlinearity introduced by ReLU disrupts the rotational symmetry of the activation space, potentially explaining why different networks converge to similar bases. In contrast, transformers use GeLU nonlinearities in MLP layers, and notably, the penultimate layer in ViTs often lacks any nonlinearity at all. These architectural choices retain greater rotational freedom in the feature space, which likely explains the lack of axis alignment across transformer runs, as also confirmed by our results. We now cite and briefly summarize this theoretical explanation in the revised manuscript (**Discussion section**) to contextualize the observed architectural difference in axis alignment between CNNs and transformers.
>
> **References**:
>
> **[1]** Privileged representational axes in biological and artificial neural networks (Khosla et al., 2024)

---

> > ### Comment · Reviewer_yuMg · 2025-08-04
> >
> > Thank you for considering and addressing the points made in the review. The extra analyses and baselines relieve my concerns regarding the technical contributions of the work, and so I will raise my score to a 5.

---

### Decision · Program_Chairs · 2025-09-17

**Decision:**

Accept (poster)

**Comment:**

This paper presents an empirical study of the phenomenon of "representational convergence" across a wide range of neural networks, training trajectories, and distribution shifts. Three alignment metrics (linear regression, orthogonal Procrustes, and permutation/soft-matching) are used across thousands of layer-pair comparisons and multiple architectures, including CNNs, ViTs, and LLMs. The main findings are that: a) orthogonal transformations align representations nearly as  effectively as more flexible linear ones, b) nearly all representational alignment seems to crystallize within the first epoch potentially pointing to shared input statistics and architectural bias playing a role on alignment vs dependence on final task, and c) for out-of-distribution images, early layers remain robustly aligned, whereas deeper layers diverge in proportion to the distribution shift.

Strengths:  The empirical scope is broad with novel insights into the timing (first-epoch convergence) and structure (orthogonal sufficiency, privileged axes) of representational alignment.  During rebuttal a number of points were addressed (e.g. sensitivity of Spearman correlation, random initialization baselines, random-pixel sampling) to alleviate reviewer concerns.

Weaknesses:
I share the concern raised by one of the reviewers about the choice of using the center pixel from convolutional feature maps as the basis for the plots. However, the rebuttal showed that via random-pixel sampling provided similar and consistent results.  There were concerns about the presentation and I hope the authors will address them in the revised draft.  The rebuttal discussion was highly engaging and positive.

Reviewer consensus:
Most reviews with the exception of the reviewer who had the methodological concern about the use of center pixel statistics were positive. I share the concern about the methodology and urge the authors to conduct further experiments to clarify their claim.

Decision::
The paper advances our understanding of representational convergence across neural networks and provides alternative viewpoints to current dialogue on representational alignment. Despite some methodological reservations, I recommend acceptance as a poster as it would be a venue for detailed discussion for the audience with the authors.